# New Body Mass Index Cut-Off Point for Obesity Diagnosis in Young Thai Adults

**DOI:** 10.3390/nu16142216

**Published:** 2024-07-11

**Authors:** Thunchanok Kuichanuan, Thamonwan Kitisatorn, Chatlert Pongchaiyakul

**Affiliations:** Department of Medicine, Khon Kaen University, Khon Kaen 40002, Thailand; thunchanok.kuichanuan@gmail.com (T.K.); monthamonwanmd@gmail.com (T.K.)

**Keywords:** obesity, BMI, body composition, body fat, normal weight obesity, NWO

## Abstract

Obesity is a global health threat affecting people of all ages, especially young adults. Early diagnosis of obesity allows for effective treatments and the prevention of its consequences. This study aimed to determine the prevalence of obesity in Thai young adults, evaluate the extent to which BMI values indicate excess adiposity, and identify the most appropriate BMI diagnostic cut-point based on body fat percentage. The study included 186 young adults aged 20 to 35 years. The diagnosis of obesity using body mass index (BMI) was compared with dual-energy X-ray absorptiometry-derived body fat percentage, considered the gold standard. The appropriate BMI cut-point was established using ROC curve analysis and the Youden index. Obesity was more common in women and in urban areas. BMI and body fat were significantly correlated; however, there was a high rate of false-negative obesity diagnosis based on the conventional BMI cut-off, a condition known as normal weight obesity (NWO). The newly proposed BMI cut-off points that best correlated with body fat and corrected false negatives were 22.1 kg/m^2^ for men and 22.5 kg/m^2^ for women. These new BMI cut-points should be applied together with clinical evaluations for obesity assessment in this particularly high-risk group.

## 1. Introduction

Obesity is a major health concern around the world, and its prevalence is increasing in both developed and developing countries. The global prevalence of obesity and overweight among persons aged 18 and older in 2016 was 13% and 39%, respectively [1,2], with one billion people globally estimated to be obese by 2030 [3]. Obesity is described by the World Health Organization (WHO) as a disorder characterized by an abnormal accumulation of adipose tissue in terms of quantity and distribution, resulting in dysfunctional adipose tissue, a chronic inflammatory state, and health impairment. Obesity-related consequences include type 2 diabetes, hypertension, dyslipidemia, fatty liver disease, hormone dysfunction, gallbladder disease, osteoarthritis, obstructive sleep apnea, and some types of cancer. Obesity is currently regarded as one of the top causes of premature death, contributing to 8% of global deaths in 2017 [1].

Obesity is commonly diagnosed in clinical practice using the body mass index (BMI). According to the WHO, BMI is a simple weight-for-height index derived by dividing body weight in kilograms by height in centimeters squared. BMI criteria for adult overweight and obesity diagnosis are equal to or greater than 25 and 30 kg/m^2^, respectively. Meanwhile, BMI criteria for overweight and obesity in adult Asians are equal to or greater than 23 kg/m^2^ and 25 kg/m^2^, respectively [4].

The BMI approach is commonly used as it is widely available, simple to use, inexpensive, repeatable, and allows for follow-up evaluations. BMI has been associated with obesity-related comorbidities such as type 2 diabetes, cardiometabolic disease, atherosclerotic cerebrovascular disease, cancer, and mortality [5,6]. Many international guidelines recommend using BMI together with clinical evaluations to screen for adiposity [6,7,8,9]. However, BMI is an indirect measure of body fat mass and may have limits in assessing adiposity, cardiometabolic risk, and mortality in certain populations. Body composition analysis, which employs bioelectrical impedance analysis (BIA) or dual-energy X-ray absorptiometry (DXA), can provide a direct measurement of total body fat mass. DXA is a non-invasive, valid, and more accurate approach to test body composition that assesses fat mass (FM), lean soft tissue or lean mass (LM), and bone mineral content, with the latter two combined as fat-free mass [10,11]. DXA is suggested for fat mass evaluation in a range of clinical settings, particularly among obese people [10,11]. The previously established gold standard measure for diagnosing obesity is body composition, with a cut-off point of body fat percentage (%BF) > 25% in men and ≥35% in women [5,6]. However, body composition measurement is not commonly used in clinical practice due to limited availability, high cost, non-bedside procedure, and greater time consumption when compared to standard care.

The present study aimed to examine the prevalence of obesity in Thai young adults, defined by %BF and BMI, determined by gender and residence (rural or urban), and to determine the degree to which the BMI value is indicative of excess adiposity, as well as to define the diagnostic cut-off point of BMI according to body fat percentage in young adult Thais to generalize in clinical practice.

## 2. Materials and Methods

### 2.1. Subjects and Setting

This study was a cross-sectional investigation conducted in Khon Kaen, Thailand. It included 186 Thai men and women aged 20 to 35 years, with 101 and 85 individuals from rural and urban areas, respectively. All participants were of Thai descent and had spent their entire lives in their respective areas without migration. This study adhered to the Helsinski Declaration of 1975 and the Good Clinical Practice Guidelines. This study was approved by the Ethical Committee of Khon Kaen University, and all participants gave written informed consent.

### 2.2. Anthropometric Measurement

Body weight (with light indoor clothing) was measured using an electronic balance (accuracy 0.1 kg), and standing height (without shoes) with a stadiometer (nearest 0.1 cm). The BMI was calculated as the ratio of weight (kg) to height (m^2^). Previously proposed WHO criteria for diagnosing Asian adult obesity were a BMI greater than 25 kg/m^2^.

### 2.3. Measures of Body Composition

Body composition, including lean mass (LM) and fat mass (FM), was determined using a DXA scanner (Prodigy, GE-LUNAR, Madison, WI, USA). The onboard software estimated FM, LM, and body fat percentage (%BF) by extrapolating fatness from the ratio of soft tissue attenuation of two X-ray energies in non-bone-containing pixels. FM and LM values were reported in kilograms. %BF was computed as the ratio of FM to body weight. Using the WHO’s suggested criteria, a man was classified as obese if his %BF was 25 or greater, while the corresponding value for women was 35. The coefficient of variance for DXA measures of body composition was between 3 and 4%.

### 2.4. Statistical Analysis

Data analysis was conducted separately for men and women, as well as in urban and rural areas. Categorical variables were reported as frequencies and percentages. The mean and standard deviation (SD) of continuous data were calculated and presented for demographic data. The mean is compared using an independent *t* test. The correlation between BMI and %BF was demonstrated using either Pearson correlation in parametric correlation or Spearman Rank correlation in nonparametric correlation. The prevalence of obesity for each gender was then determined. Simple and multiple Linear regression was used to estimate the effect of each variable on %BF. McNemar chi-square cross-tabulation was used to compare the prevalence of obesity based on BMI and %BF.

To establish the optimal diagnostic cut-off for BMI as a proxy measure of obesity, a number of receiver operating characteristic (ROC) curves were created with %BF as the gold standard. In this study, the Youden index (Sensitivity + Specificity − 1) was used to determine the ROC data cut-off point. All data were analyzed with IBM SPSS Statistics 26, which was provided by Khon Kaen University.

### 2.5. Sample Size Calculation

Buderer’s formula was used to calculate the sample size, with an alpha (α) of 0.05 and an absolute precision of 0.12 (d = 12%). The prevalence of obesity in Thai adults was 42.2% [12]. According to Okorodudu DO et al. [13], the sensitivity and specificity of BMI for obesity diagnosis were 0.5 (95% confidence interval (CI): 0.43–0.57) and 0.9 (CI: 0.86–0.94), respectively, with an expected dropout rate of 5%. Therefore, the final sample size was estimated to be 168.

## 3. Results

A total of 186 participants (90 men and 96 women) were recruited in this study. Age and BMI were comparable between the sexes; however, men had significantly higher body weight, height, LM, and %LM, while they had lower FM and %BF than women. Table 1 shows that rural participants had significantly higher BMI and %LM, but lower FM and %BF than the urban participants.

There were moderately positive relationships between BMI and %BF in both men and women, with a correlation coefficient (r) of 0.556 (*p* < 0.01) in men and 0.530 (*p* < 0.01) in women. Urban men had the highest positive correlation between BMI and %BF (r = 0.784, *p*< 0.01), whereas other subgroups had moderately positive correlations (r) of 0.590 (*p* < 0.01) in rural men, 0.532 (*p* < 0.01) in urban women, and 0.630 (*p* < 0.01) in rural women.

When compared to rural men, urban men had significantly higher body weight and height, but similar BMI. However, in urban men, FM and %BF were significantly higher, while %LM was significantly lower than in rural men. Body weight, BMI, LM, and FM were significantly lower in urban women than in rural women, while %BF was comparable (Table 2).

Table 3 shows the prevalence of obesity in men and women residing in rural and urban settings, as measured by BMI and %BF. In young men and women, the overall prevalence of obesity was 6.7% and 17.7%, respectively. When both BMI and %BF criteria were taken into account, the prevalence of obesity in men was higher in urban participants than in rural individuals. In women, obesity was more common in rural areas than in urban areas, with 28.3% and 22.6% of rural women obese based on BMI and %BF, respectively.

In order to determine the diagnostic potential of BMI when using %BF as a gold standard for obesity diagnosis, the data were shown with a scatter plot (Figure 1). We found that 52.2% of all obese individuals met BMI criteria, and 47.8% were classified non-obese as false negatives. A total of 33.3% of obese men and 58.8% of obese women were classified as obese using BMI criteria, respectively. The current study found that the majority of non-obese individuals (93.3%) were classified as non-obese using BMI criteria, whereas 6.7% were classified obese as false positives.

To determine the optimal BMI cut-offs for diagnosis of obesity in young men and women, ROC curve analysis and the Youden index were performed (Figure 2). The highest Youden index associated with BMI cut-offs was 22.1 kg/m^2^ for men and 22.5 kg/m^2^ for women, with AUCs of 0.850 (95% CI: 0.727–0.973) and 0.812 (95% CI: 0.689–0.934), respectively. Table 4 shows the diagnostic performance for obesity based on the WHO’s BMI criteria and proposed BMI cut-offs. Using BMI levels of ≥25 kg/m^2^, the prevalence of obesity was 6.7% and 17.7% in men and women, respectively. The sensitivity of the BMI WHO criteria was low (33.3% in men and 58.8% in women), even with a high specificity of 91.1–95.2%. However, using the present study’s proposed cut-off criteria (BMI ≥ 22.1 kg/m^2^ in men and ≥22.5 in women), the sensitivity of the proposed criteria increased to 100% and 82.4% in men and women, respectively, while the specificity decreased to 60.7% in men and 72.2% in women. With the newly proposed BMI cut-off, false positive rates were 39.3% and 27.8% in men and women, respectively, while false positive rates by residences were 33.3% for urban men, 43.8% for rural men, 13.2% for urban women, and 41.5% for rural women.

## 4. Discussion

The present study focused on obesity diagnosis in young Thai adults, specifically the prevalence of obesity by sex and residence, and compared diagnostic approaches using body composition analysis, %BF versus BMI. The population was entirely Asian, which is well documented to differ from Western populations in terms of how body fat or BMI affects metabolic and cardiovascular risk. Furthermore, the current study contributes to research efforts by using the DXA test to assess %BF, which is more accurate in diagnosing obesity.

A previous study found that Asians had a 3–5% higher body fat percentage than Caucasians at the same BMI, and with the same %BF, Asians were 3–4 kg/m^2^ lower in BMI [14]. Studies have reported that Asians have a higher prevalence of metabolic complications and cardiovascular risks, such as type 2 diabetes and hypertension, than Caucasians with the same BMI [15,16,17]. Another study in Vietnam found that using BMI instead of DXA-derived %BF, with obesity diagnosis at %BF > 30% in males and >40% in females, led to the underdiagnosis of obesity [18]. Because of the lack of specific cut-off points for different ethnicities regarding adverse outcomes related to obesity or cardiovascular issues in Thai young adults, we utilized %BF as the standard for diagnosing obesity, with a threshold of ≥25% in men and ≥35% in women, as recommended by the WHO and the 2016 AACE/ACE obesity clinical practice guidelines [5,6]. The fat mass index, calculated by dividing the total body fat mass (in kilograms) by the square of the height (in meters), is another way to evaluate adiposity [6], although its usefulness is restricted by limited ethnic-specific thresholds.

The study showed differences in anthropometric and body composition characteristics among young people based on sex and residence area. When %BF was used as a diagnostic tool, young women tended to have higher levels of obesity compared to men, a trend consistent with other global studies [19]. When using BMI as the diagnostic tool, there was no difference between the two sexes. Obesity is more common in women than men for several reasons, including societal, environmental, behavioral, and physiological factors, metabolic responses, and hormonal impacts [20]. Men and women have distinct body compositions and fat distribution patterns. Men generally have a greater proportion of fat-free mass, whereas women typically have more body fat compared to men [21]. Women typically have around 10% more body fat than men at the same BMI [22]. BMI is unable to distinguish between fat mass and fat-free mass, leading to a potential underestimation of body fat percentage in women and an overestimation in men [19]. The reported relative risk of obesity-related metabolic complications, such as type 2 diabetes, hypertension, and coronary artery disease, was higher in overweight and obese patients (BMI > 25 and 30 kg/m^2^, respectively) compared to healthy weight participants. This difference was more pronounced in women than in men [23]. There is also a differential in fat distribution patterns across sexes, which is due to hormonal effects. In women, estrogen and progesterone increase overall adiposity, which is distributed peripherally. While men have higher levels of testosterone, at the same BMI, they have more visceral adiposity and ectopic adipose tissue deposition, including intrahepatic and intramyocellular lipid, which has a negative impact on insulin sensitivity and cardiometabolic risk [19,20,21,24]. As a result, many obesity guidelines recommend waist circumference as the most practicable tool for assessing abdominal fat [6,9,25,26,27].

We found that urban young adults had a higher %BF and a lower %LM than those who lived in rural areas. In contrast, urban living is associated with a lower BMI. Other studies that used BMI as a comparison method found an inconsistent relationship between urban or rural life and obesity prevalence, which varied by the study population [12,28,29,30,31]. Obesity prevalence, however, was consistently increasing in both rural and urban settings across all investigations. There was no clear trend across sexes based on their residences; urban men were more likely to be obese in terms of body composition and had less lean mass than rural men. Urban women had lower BMI but similar body composition.

We observed a moderate to high degree of positive correlation using two diagnostic approaches. The correlation in urban men (r = 0.784, r < 0.01) was the strongest among the four participant categories. There is a strong correlation between BMI and %BF, as BMI primarily indicates fatness in populations where muscle mass is neither exceedingly high nor low. The correlation was less extent in rural men who were higher in lean mass compared with other groups. However, we do not anticipate a significantly altered body composition pattern in general young adults; therefore, the BMI may serve as a useful indicator of body fat and a diagnostic tool for obesity in this age group.

In this study, we found that the prevalence of obesity using the BMI method was equivalent to that of the gold standard %BF; nevertheless, the BMI method produced a significant proportion of false positives and false negatives. A false negative in BMI refers to individuals with a normal BMI but excessive body fat, also known as normal weight obesity (NWO). This is a critical issue because it can lead to a delayed diagnosis, inadequate health concern on the part of both patients and healthcare providers, and subsequent postponed intervention, all of which have the potential to negatively affect an individual’s health. Furthermore, findings from a previous study indicated that young adults diagnosed with NWO who had a healthy BMI ranging from 18.5 to 24.9 kg/m^2^ but an excessive %BF (>23% in men and >30% in women) had a significantly higher risk of metabolic syndrome, characterized by increased waist circumference, elevated triglycerides, decreased HDL-cholesterol levels, and increased insulin resistance [32]. A previous investigation demonstrated that %BF performed a more accurate prognostic function in predicting cardiovascular risk factors than BMI [33]. Cardiovascular risk factors were 1.88 times more prevalent in the NWO group compared to the lean group, whereas inappropriate BMI in the normal %BF group was not associated with an increase in risk, and %BF was independently associated with cardiovascular risk factors [33]. The findings of our study indicate that the prevalence of NWO among young adults in Thailand was 47.8%. Among these individuals, 66.7% were obese men and 41.2% were obese women. These high percentages should raise concern. Our prevalence was similar to a previous study on Thai young female adults which demonstrated a 46.8% prevalence of NWO [34]. The prevalence of NWO differs among ethnicities and diagnostic criteria [35]. In Koreans, NWO prevalence of 36% in men and 29% in women was found [36]. Among North Americans, the overall NWO prevalence was 33.4% [37]. Young adults are particularly vulnerable to weight gain when compared to senior adults, which increases the likelihood that they will develop obesity, particularly among those who were initially overweight or obese as young adults [38,39,40]. The mean annual weight gain for young adults in developing countries was 1 kg [41,42], while in developed countries it ranged from 0.4 to 0.9 kg [38,43].

Obesity in young people is a predictor of adult obesity, and it frequently continues and increases as they become older [44]. Obese young people showed a concurrent shift in cardiometabolic risk factors [41,45,46,47]. Significant lifestyle changes, reduced physical activity, higher sedentariness, urbanization, nutritional transition, eating disorders, depression, sleep disturbance, and sociocultural shifts are all risk factors for weight gain in young adults [39,41]. Early identification of obesity and NWO in young adults would be critical for preventing adult obesity. This approach demands a correct method of diagnosing obesity and NWO in this specific demographic. Body composition analysis can discriminate between fitness and fatness; however, it is often expensive, time-consuming, and limited in availability. The approach proposed in this study involves using a BMI cut-off point, which is largely related to body fat percentage. We found a new diagnostic BMI cut-off point of 22.1 kg/m^2^ for men and 22.5 kg/m^2^ for women, which significantly improved diagnostic sensitivity. Even though these new cut-off points are similar to the WHO criteria for overweight Asian [4], we propose that those with a BMI above these new cut-off points be classified as “at risk of obesity” rather than simply “overweight”.

By proposing the new BMI cut-off point, we aimed to raise the health concern of obesity in this population. Obesity is generally preventable, costs less to treat at an early stage, and has a low risk of treatment-related adverse effects. A screening test for such a disease should be very sensitive and have a low false negative rate. We propose using a new cut-off point for early diagnosis and intervention, which is the mainstay of adult obesity prevention and management while also lowering the worldwide obesity-related health burden [1]. With the lowered cut-off point, the false positive rate of obesity diagnosis may increase, particularly in men and rural residents who were higher in lean mass. For our pragmatic recommendation, BMI can still be used in conjunction with clinical evaluations of body composition, including muscle mass, volume status, and body fat, particularly waist circumference, with ethnic-specific cutoffs for visceral fat tissue estimation, which can provide information comparable to body composition analysis in limited availability settings [48]. However, given the available scenario, body composition analysis may provide additional information about body fat quantity and distribution in people with equivocal adiposity status. Furthermore, the new BMI cut-off point should be addressed while dealing with the NWO’s general young adult population. Our findings are mostly consistent with earlier studies [49,50,51,52]; however, the current study is the first to propose a novel diagnostic cut-off point for Asian young adults that is most closely associated with %BF. Furthermore, DXA-derived body composition analysis is widely acknowledged as the gold standard for determining body fat, particularly in obese individuals [10,11].

The present findings must be interpreted in the context of a number of strengths and limitations. To our knowledge, this was the first study to investigate young Thai adults, and the measurement of body fat and fat-free mass in this study was measured using DXA machines, which is considered one of the most accurate and valid techniques for measurement. However, it is crucial to note that our study had some limitations. Firstly, the study participants were Thai, whose body sizes, lifestyles, cultural backgrounds, and environmental living conditions are different from other populations. Thus, care should be taken when extrapolating these results to other populations. Secondly, the study excluded data on additional metabolic parameters such as waist circumference, visceral and subcutaneous adipose tissue differentiation, obesity-related complications/comorbidities, metabolic and cardiovascular outcomes, and physical activity level. Further research including these data would be valuable in ensuring the therapeutic use of these new cut-off points. Third, our study population consisted of 186 participants aged 20 to 35. Although this followed the statistical procedure, the sample size was small, and the age range was narrow. A larger population-based investigation would be required to confirm this new cut-off point and extend the findings to all Thai populations. Lastly, the optimal clinically significant %BF for predicting cardiometabolic adverse outcomes in the Thai population remains unknown, requiring further high-quality prospective research studies.

## 5. Conclusions

Young adults are at high risk of becoming obese, and there is a high rate of normal weight obesity (NWO), which has a deleterious impact on their metabolic profile. With the standard obesity diagnostic cut-off point, a large number of obese patients are underdiagnosed, resulting in delayed treatment and consequence prevention. Our newly proposed obesity diagnosis cut-off point of 22.1 kg/m^2^ for men and 22.5 kg/m^2^ for women greatly improves diagnostic sensitivity, potentially leading to early detection and primary prevention of the obesity-related morbidity and mortality burden.

## Figures and Tables

**Figure 1 nutrients-16-02216-f001:**
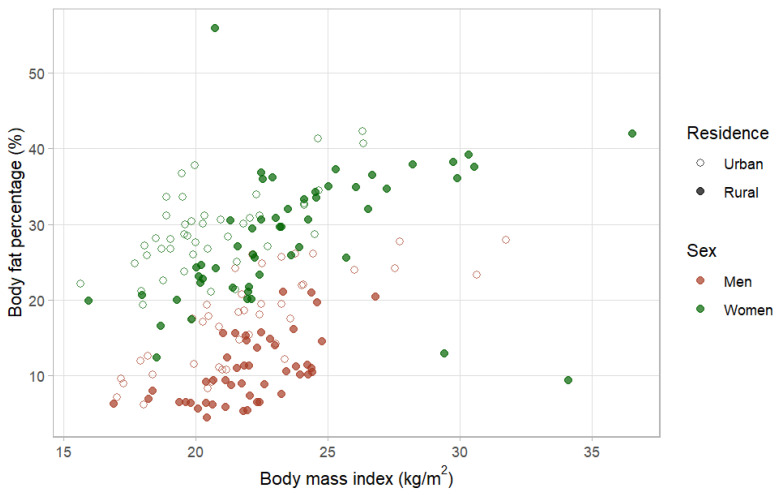
Scatter plot of body fat percentage and BMI regarding residence and in young men and women.

**Figure 2 nutrients-16-02216-f002:**
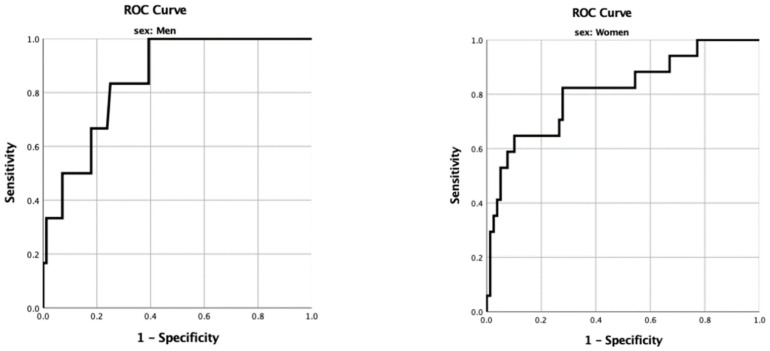
ROC curve of %BF and BMI in young men and women.

**Table 1 nutrients-16-02216-t001:** Demographic characteristics of study participants.

All	By Sex		By Living Area	
	Men (N = 90)	Woman (N = 96)	Rural (N = 101)	Urban (N = 85)
Age (year)	27.7 ± 4.6	27.8 ± 4.0	27.9 ± 4.4	27.5 ± 4.1
Body weight (kg)	61.2 ± 9.0 *	53.7 ± 9.3	57.2 ± 9.1	57.5 ± 10.7
Height (cm)	166.6 ± 6.4 *	155.2 ± 5.3	158.1 ± 7.5	163.8 ± 7.8 *
BMI (kg/m^2^)	22.0 ± 2.6	22.3 ± 3.7	22.9 ± 3.3 **	21.3 ± 2.9
Lean mass (kg)	48.9 ± 5.3 *	34.3 ± 4.2	42.1 ± 8.1	40.6 ± 9.4
Fat mass (kg)	8.8 ± 5.2	15.7 ± 5.8 *	11.5 ± 7.2	13.4 ± 5.4 **
% Lean mass (%)	85.4 ± 6.7 *	69.3 ± 6.3	78.9 ± 11.4 **	75.0 ± 9.0
% Body fat (%)	13.8 ± 6.3	28.8 ± 7.3 *	20.0 ± 11.2	23.4 ± 8.4 **

* *p* < 0.001, ** *p*< 0.05.

**Table 2 nutrients-16-02216-t002:** Demographic characteristics of study participants by sex and residence.

All	Men		Women	
	Rural (N = 48)	Urban (N = 42)	Rural (N = 53)	Urban (N = 43)
Age (year)	27.8 ± 4.7	27.6 ± 4.4	28.1 ± 4.2	27.3 ± 3.8
Body weight (kg)	59.1 ± 6.2	63.7 ± 11.0 **	55.5 ± 10.9	51.5 ± 6.2 **
Height (cm)	163.8 ± 5.8	169.8 ± 5.6 *	153.0 ± 4.6	158.0 ± 4.8 *
BMI (kg/m^2^)	22.0 ± 1.9	22.1 ± 3.2	23.7 ± 4.0 *	20.6 ± 2.4
Lean mass (kg)	49.1 ± 4.4	48.6 ± 6.3	35.7 ± 4.4 *	32.7 ± 3.3
Fat mass (kg)	6.4 ± 3.1	11.5 ± 5.8 *	16.0 ± 6.8 *	15.3 ± 4.4
% Lean mass (%)	88.7 ± 4.7 *	81.7 ± 6.7	70.0 ± 7.7	68.6 ± 5.7
% Body fat (%)	10.7 ± 4.5	17.3 ± 6.3*	28.4 ± 8.6	29.3 ± 5.4

* *p* < 0.001, ** *p*< 0.05.

**Table 3 nutrients-16-02216-t003:** Comparison of prevalence of obesity using BMI and %BF by sex and living area.

		BMI ≥ 25 kg/m^2^	% Body Fat (Obesity: Men ≥ 25%, Women ≥ 35%)
Men	Rural	2.1% (1/48)	0% (0/48)
	Urban	11.9% (5/42)	14.3% (6/42)
	All	6.7% (6/90)	6.7% (6/90)
Woman	Rural	28.3% (15/53)	22.6% (12/53)
	Urban	4.7% (2/43)	11.6% (5/43)
	All	17.7% (17/96)	17.7% (17/96)
All		12.4% (23/186)	12.4% (23/186)

**Table 4 nutrients-16-02216-t004:** Cut-off values for body mass index for the diagnosis of obesity (%BF ≥ 25% for men and ≥35% for women) and associated diagnostic indices.

	Men	Women
**WHO cut-off**	**≥25 kg/m^2^**	**≥25 kg/m^2^**
Prevalence	6.7% (2.5–13.9)	17.7% (11.0–26.8)
Sensitivity	33.3% (4.3–77.7)	58.8% (32.9–81.6)
Specificity	95.2% (88.3–98.7)	91.1% (82.6–96.4)
Positive predictive value	33.3% (4.3–77.7)	58.8% (32.9–81.6)
Negative predictive value	95.2% (88.3–98.7)	91.1% (82.6–96.4)
Positive likelihood ratio	7.0 (1.6–30.8)	6.6 (3.0–14.9)
Negative likelihood ratio	0.7 (0.4–1.2)	0.5 (0.3–0.8)
**Proposed cut-off**	**≥22.1 kg/m^2^**	**≥22.5 kg/m^2^**
Sensitivity	100.0% (54.1–100.0)	82.4% (56.6–96.2)
Specificity	60.7% (49.5–71.2)	72.2% (60.9–81.7)
Positive predictive value	15.4% (5.9–30.5)	38.9% (23.1–56.5)
Negative predictive value	100.0% (93.0–100.0)	95.0% (86.1–99.0)
Positive likelihood ratio	2.6 (2.0–3.3)	3.0 (2.0–4.5)
Negative likelihood ratio	0	0.2 (0.1–0.7)

Values: % (95% CI).

## Data Availability

The data presented in the study are available on reasonable request from the corresponding author due to ethical reason.

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
