# Peer review of "New Body Mass Index Cut-Off Point for Obesity Diagnosis in Young Thai Adults"

_nutrients, 2024, doi:10.3390/nu16142216_

Round 1
Reviewer 1 Report
Comments and Suggestions for Authors
The study by Kuichanuan et al. demonstrated that by comparing BMI with % body fat determined using DEXA there was a considerable number of individuals who would be considered obese that were not detected using the BMI alone. Using their results that they have presented, the authors argue that BMI cut-offs for obesity should be lowered to capture the group they describe as "false negatives" who fall under the non-obese BMI range but would be considered obese using DEXA % body fat. This study is of interest and importance in the field of obesity and adds to the important discussion of BMI cut-offs in various ethnic groups. I have no major changes to recommend that I feel would be of use to the authors, however, there are some minor recommendations that I have listed below which I think the authors should consider:
Last row of table 3: 188 should be 186
Lines 139-143 – It would be helpful for the reader if you could depict this data in a graph for example with dots representing each individual of BMI vs %BF (different shapes for men/women and different colours for rural/urban). This would allow the reader to see the proportion of false negatives and false positives at different cut-off points. This is an important point you should highlight.
It is important for the authors to present in the results and discuss how many false positives would come up if BMI cut-offs were to be lowered to their suggested ones, and what proportion of those false positives would be rural and urban, men and women.
BMI has limitations when it comes to individuals with greater % lean mass/muscle mass which is the case for rural populations. Would the new BMI cut-offs that the authors suggest give a high rate of false positives in this population group? The authors should discuss this point and its possible impact. Might a high false positive rate cause people to take an obese BMI less seriously?
The authors mention that with the current recommended BMI cut-offs of 23 for obesity there are many false negatives, how do the rates of false negatives compare with other ethnicities?
Overall, I commend that authors on the interesting study that was presented to a high standard.
Author Response
Dear Editor Board Member,
Nutrients
Thank you very much for your letter dated June 26, 2024 together with the reviewer’s comments
our manuscript. We very much appreciate the thoughtful and helpful review of our work, and
thank the reviewer for the suggestions to enhance the manuscript. We have carefully considered
the comments and would like to take the opportunity to address to the comments in the following
attachment. Please see the attached for our point-by-point response. Changes are highlighted in
red in the revised manuscript.
We consider that the manuscript has improved significantly as a result of the reviewer’s
comments. We hope that the revised manuscript is satisfactory to you, and is possible for
publication in the Nutrients
.
Your Sincerely,
Chatlert Pongchaiyakul
Division of Endocrinology and Metabolism
Department of Medicine, Faculty of Medicine
Khon Kaen University, THAILAND 40002.
Tel: +66-43-363664
Email: pchatl@kku.ac.th
Corresponding author
------------------------------------------------------------------------------------------------------------Comment 1 : Lines 139-143 – It would be helpful for the reader if you could depict this data in a graph for example with dots representing each individual of BMI vs %BF (different shapes for men/women and different colours for rural/urban). This would allow the reader to see the proportion of false negatives and false positives at different cut-off points. This is an important point you should highlight.
Response : Thank you for the suggestion. We prepared the scatter plot as suggested by the reviewer in figure 1.
Comment 2 : It is important for the authors to present in the results and discuss how many false positives would come up if BMI cut-offs were to be lowered to their suggested ones, and what proportion of those false positives would be rural and urban, men and women.
Response : Thank you for the suggestion, We reported the false positive rate of the revised BMI cut point for each subgroup in the paragraph (Lines 159-162) and discussed it (Lines 284-285).
Comment 3 : BMI has limitations when it comes to individuals with greater % lean mass/muscle mass which is the case for rural populations. Would the new BMI cut-offs that the authors suggest give a high rate of false positives in this population group? The authors should discuss this point and its possible impact. Might a high false positive rate cause people to take an obese BMI less seriously?
Response : Thank you for the suggestion. We totally agree that BMI alone cannot distinguish between lean and fat mass; nonetheless, it is well known that obesity is generally preventable, that early treatment is less expensive, and that treatment-related adverse effects are modest. A screening test for such a disease should be highly sensitive with a low false-negative rate. We recommend utilizing a new cut-off point in conjunction with clinical body composition evaluations, such as anthropometric measurement or body composition analysis, in available settings for early and accurate obesity diagnosis and intervention.
Comment 4 : The authors mention that with the current recommended BMI cut-offs of 23 for obesity there are many false negatives, how do the rates of false negatives compare with other ethnicities?
Response: Thank you for the suggestion. We have added the information about this point in the paragraph (Line 254-258).

Reviewer 2 Report
Comments and Suggestions for Authors
I've received the paper "New BMI cut-point for obesity diagnosis in young Thai adults" congratulations on your work. The study aims to examine the prevalence of obesity in rural or urban Thai young adults, defined by % BF and BMI, and to define diagnostic cut-point of BMI according to body fat percentage in young adult Thais to generalize in clinical practice. The research offers interesting contributions regarding the BMI. However, there are significant criticisms that I would like to be clarified.
General Comments: The manuscript is interesting and well written, however presents some areas of weakness, including size and inclusion criteria of the sample.
In my opinion, the BMI is not an adequate parameter for diagnosing overweight and obesity in all population groups, the amount of physical activity and exercise are variables that affect both fat and muscle weight in people, as well as in their percentages in individuals. As the authors point out in lines 193-194, BMI does not take these parameters into account. Therefore, it is necessary to consider this variable in the study in a heterogeneous group with the same physical activity and amount of exercise.
Furthermore, the study presents a significant limitation, as the sample consists of only 186 subjects aged between 20 and 35 years. New cut-off points are proposed generalize for an age range in which currently several million people live in Thailand.
In this form, I believe it is not considered acceptable for publication, but I give the authors the opportunity to reflect on my concerns..
Author Response
Dear Editor Board Member,
Nutrients
Thank you very much for your letter dated June 26, 2024 together with the reviewer’s comments
our manuscript. We very much appreciate the thoughtful and helpful review of our work, and
thank the reviewer for the suggestions to enhance the manuscript. We have carefully considered
the comments and would like to take the opportunity to address to the comments in the following
attachment. Please see the attached for our point-by-point response. Changes are highlighted in
red in the revised manuscript.
We consider that the manuscript has improved significantly as a result of the reviewer’s
comments. We hope that the revised manuscript is satisfactory to you, and is possible for
publication in the Nutrients
Your Sincerely,
Chatlert Pongchaiyakul
Division of Endocrinology and Metabolism
Department of Medicine, Faculty of Medicine
Khon Kaen University, THAILAND 40002.
Tel: +66-43-363664
Email: pchatl@kku.ac.th
Corresponding author
------------------------------------------------------------------------------------------------------------
Comment 1 : In my opinion, the BMI is not an adequate parameter for diagnosing overweight and obesity in all population groups, the amount of physical activity and exercise are variables that affect both fat and muscle weight in people, as well as in their percentages in individuals. As the authors point out in lines 193-194, BMI does not take these parameters into account. Therefore, it is necessary to consider this variable in the study in a heterogeneous group with the same physical activity and amount of exercise.
Response : Thank you for the suggestion. We completely agree with your statement. Exercise and physical activity affect muscle mass, as well as body weight and BMI. We accept this issue as a study limitation and include it in the discussion (Lines 308-309). We believe that additional high-quality research using these types of data is necessary and would be useful.
Comment 2 : Furthermore, the study presents a significant limitation, as the sample consists of only 186 subjects aged between 20 and 35 years. New cut-off points are proposed generalize for an age range in which currently several million people live in Thailand.
Response : Thank you for the suggestion. According to our study, sample size is calculated using the standard equation and the referential prevalence from the landmark study. We understand your concern and believe that a population-based study would be more reliable; unfortunately, this type of data in the Thai population was not previously available. We additionally think that our study methodology and sample size calculation are standardized, and that the findings are valuable and generalizable.

Round 2
Reviewer 2 Report
Comments and Suggestions for Authors
The authors have resolved my concerns.
Author Response
Dear Editor Board Member,
Nutrients
Thank you very much for all the valuable comments. All of us in the author team always appreciate your suggestion and we have revised our work to be a better version as you suggested.
We hope that the revised manuscript is proper for publication in the Nutrients
.
Your Sincerely,
Chatlert Pongchaiyakul
Division of Endocrinology and Metabolism
Department of Medicine, Faculty of Medicine
Khon Kaen University, THAILAND 40002.
Tel: +66-43-363664
Email: pchatl@kku.ac.th
Corresponding author